# Generic Structure Extraction with Bi-Level Optimization for Graph Structure Learning

**DOI:** 10.3390/e24091228

**Published:** 2022-09-01

**Authors:** Nan Yin, Zhigang Luo

**Affiliations:** School of Computing, National University of Defense Technology, Changsha 410000, China

**Keywords:** graph neural network, graph structure learning, noise learning, bi-level optimization

## Abstract

Currently, most Graph Structure Learning (GSL) methods, as a means of learning graph structure, improve the robustness of GNN merely from a local view by considering the local information related to each edge and indiscriminately applying the mechanism across edges, which may suffer from the local structure heterogeneity of the graph (i.e., the uneven distribution of inter-class connections over nodes). To overcome the drawbacks, we extract the graph structure as a learnable parameter and jointly learn the structure and common parameters of GNN from the global view. Excitingly, the common parameters contain the global information for nodes features mapping, which is also crucial for structure optimization (i.e., optimizing the structure relies on global mapping information). Mathematically, we apply a generic structure extractor to abstract the graph structure and transform GNNs in the form of learning structure and common parameters. Then, we model the learning process as a novel bi-level optimization, i.e., *Generic Structure Extraction with Bi-level Optimization for Graph Structure Learning (GSEBO)*, which optimizes GNN parameters in the upper level to obtain the global mapping information and graph structure is optimized in the lower level with the global information learned from the upper level. We instantiate the proposed GSEBO on classical GNNs and compare it with the state-of-the-art GSL methods. Extensive experiments validate the effectiveness of the proposed GSEBO on four real-world datasets.

## 1. Introduction

Based on the *homophily assumption* of “like to associate with like” [1,2], the Graph Neural Network (GNN) [3] has become the promising solution for node classification. However, a large portion of edges are inter-class connections [4], and representation propagation over such connections can largely hinder the GNN from obtaining class-separated node representations, hurting the performance.

Existing GSL methods are roughly categorized into the attentive mechanism, noise detection, and probabilistic mechanism. ***Attentive mechanism*** calculates weights for edges to adjust the contribution of different neighbors during representation propagation [5,6,7,8,9]. These methods can hardly work well in practice for two reasons: (1) the mechanism may not generalize well to all nodes with different local structures (*cf.*
Figure 1a); and (2) the attention cannot be easily trained well due to the limited labeled data [10]. ***Noise detection*** incorporates an edge classifier to estimate the probability of inter-class connection for each edge [11,12,13,14,15,16]. Although it can be better trained owing to the supervision of edge labels, the edge classifier also suffers from local structure heterogeneity and lacks consideration of global information. ***Probabilistic mechanism*** models connection from a global view, which assumes a prior distribution of edge and estimates GNN parameters with Bayesian optimization to overcome the impact of inter-class edges [17,18,19,20]. Although the edge specific parameterization largely enhances the model representation ability, it is hard to accurately access the prior distribution.

Despite the achievements of the existing methods, there still exists some common drawbacks: (1) **Edge modeling method,** the existing methods model edges with the parameter sharing mechanism, which may suffer from the local structure heterogeneity problem; (2) **Local optimization**, the local optimization problem focuses on optimizing the parameters with the information of neighbor nodes, which ignores the impact from the global view. Therefore, we come up with the key considerations for GSL: (1) modeling graph connection in an edge-specific manner instead of a shared mechanism; and (2) optimizing the corresponding parameters with a global objective of accurately classifying all nodes. The edge-specific modeling can overcome the local structure heterogeneity, i.e., handling nodes with different properties (e.g., *node 1* and *node 9* in Figure 1a) via different strategies. Besides, blindly removing the inter-class edges will increase the risk of misclassifying the target nodes (in dash circle) due to cutting off their connections to the labeled neighbors in the same class (e.g., edge between *node 5* and *node 9* in Figure 1b). Thus, it is necessary to optimize the graph structure from the global view instead of the local ones.

However, it is non-trivial to achieve the targets mentioned above due to the following challenges: (1) the graph structure is embedded into the GNN model, which affects the procedure of model parameter optimization once updated, requiring a careful design of the model training algorithm; (2) the calculation of the ideal global objective is intractable due to the limited labelled nodes, especially the hard semi-supervised setting.

In this work, we propose a new *Generic Structure Extraction with Bi-level Optimization for Graph Structure Learning (GSEBO)*, which optimized the graph structure and learsn the node embeddings from the global view. In particular, we first devise a new generic structure extractor, which accounts for the graph structure with both the connectedness between nodes and the strength of connections. In addition to the adjacency matrix, the extractor adopts a learnable matrix to represent the graph structure and adjusts the representation propagation. Moreover, we design a bi-level optimization algorithm where the outer and inner optimizations update the structure and the parameters of the base graph convolution (vs feature mapping parameters). In this way, we decompose the hard optimization issue of GSEBO into two easy ones. In addition, we separate the training set into two parts (e.g., train_1, train_2), and set the objective of outer optimization as the train_2 loss to better approximate the ideal global objective. The proposed generic structure extractor can be extended to most existing graph convolution operators. We instantiate it on four representative GNN models (i.e., GCN [21], GAT [5], GraphSAGE [22], and JK-Net [23]) and compare GSEBO with state-of-the-art GSL methods, which are evaluated on four real-world node classification datasets. Extensive experiments justify the rationality, effectiveness and robustness of the proposed method. In summary, our main contributions are as follows:We propose a novel *GSEBO* with bi-level optimization for edge-specific graph structure learning, which learns the graph structure from a global view by optimizing a global objective of node classification.We devise a generic structure extractor, which parameterizes the strength of each edge during representation propagation. Besides, we summarize how the classical GNN methods are transferred in the form of a learnable graph structure with a generic structure extractor.We evaluate the proposed GSEBO with classical GNNs as backbones and compare it with the state-of-the-art GSL methods. Extensive experiments on four real-world datasets show the superior learning ability of the proposed method compared to the existing methods.

## 2. Related Work

### 2.1. Attentive Mechanism

The attentive mechanism methods adaptively learn the weights of edges and adjust the contributions of neighbor nodes. MAGNA [6] incorporates multi-hop context information into every layer of attention computation. IDGL [8] uses the multi-head self-attention mechanism to reconstruct the graph, which has the ability to add new nodes without retraining. HGSL [9] extends the graph structure learning to heterogeneous graphs, which constructs different feature propagation graphs and fuses these graphs together in an attentive manner. However, those methods suffer from different local structures and are difficult to train.

### 2.2. Noise Detection

The noise detection methods leverage the off-shelf pre-trained model to induce node embeddings or labels and incorporate an edge classifier to estimate the probability of each edge. NeuralSparse [11] considers the graph sparsification task by removing irrelevant edges. GAUG [12] utilizes a GNN to parameterize the categorical distribution instead of MLP in NerualSparse. PTDNet [13] prunes task-irrelevant edges by penalizing the number of edges in the sparsified graph with parameterized networks. Even though the noise detection methods can be well trained with the supervision of edge labels, the edge classifier also suffers from local structure heterogeneity and lacks consideration of global information.

### 2.3. Probabilistic Mechanism

This type of method assumes the prior distribution of graph or noise and estimates the parameters through observed values, then resamples the edges or noise to obtain a new graph. BGCN [17] estimates the parameter distribution of edges and communities by sampling edges from graph, and resamples new graphs with the estimated parameters for prediction. VGCN [18] trains a graph distribution parameter similar to the original structure through ELBO, and resamples graphs for prediction. However, both the BGCN and VGCN models are sampled from a noisy graph, and the estimated parameters also contain noise. DenNE [19] assumes the observed graph is composed of real values and noise and the prior distribution of features and that the noise is known. With a generative model, the likelihood is used to estimate the representation of nodes. However, this method highly relies on the priors of feature and noise, which is difficult to obtain accurately.

### 2.4. Bi-Level Optimization on GNN

LDS [15] jointly learns the graph structure and GNN parameters by solving a bi-level optimization issue that learns a discrete probability distribution for each edges. According to the learned distributions, LDS generates a new graph structure by sampling. Towards this end, LDS sets the objective of outer optimization as generating the observed edges, which clearly has a gap to the overall classification objective. Moreover, LDS needs to estimate $N^{2}$ distribution parameters at least, which is hard due to insufficient labels ($\left|E\right|\ll N^{2}|$. On the contrary, our method only activates a small portion of entries in Z, where $A_{ij}=1$, i.e., the number of estimated parameters is same as the number of edges in graph G (i.e., |E|).

## 3. Methodology

We first introduce the essential preliminaries for GNN, and then elaborate the *graph convolution operator* and *bi-level optimization algorithm* of the proposed GSEBO.

### 3.1. Preliminary

Let G=(V,E,X) represents a graph with *N* nodes and *M* edges, where $V=\{v_{1},v_{2},\cdots ,v_{N}\}$ and $E=\{e_{1},e_{2},\cdots ,e_{M}\}$ denote the set of nodes and edges, respectively. $X={\left[x_{1},x_{2},\cdots ,x_{N}\right]}^{⊤}\in R^{N\times C}$ are nodes features, where $x_{i}\in R^{C}$ is the *i*-th row of X, corresponds to node $v_{i}$ in the form of a *C*-dimensional vector. The adjacency matrix $A\in {\left\{0,1\right\}}^{N\times N}$ indicates the connectedness of node pairs.

This task aims to learn a classifier f(A,X;θ) from a set of labeled nodes to forecast the remaining nodes labels, where θ denotes model parameters. Assuming there are *N* labels, we index them from 1 to *N* without loss of generality. Formally, $Y={\left[y_{1},y_{2},\cdots ,y_{N}\right]}^{⊤}\in R^{N}$ are the labels of the nodes, where $y_{i}\in R$ is the label of node *i*. The target is achieved by optimizing the model parameter θ with respect to the labeled nodes, which is formulated as:(1)$$\begin{array}{c} \min_{\theta }\sum_{i\leq M}l\left(f{\left(A,X\right)}_{i},y_{i};\theta \right)+\lambda ∥\theta ∥, \end{array}$$
where l(·) is a classification loss and λ is a hyperparameter to adjust the strength of parameter regularization.

### 3.2. Generic Structure Extraction

To optimize the graph structure, the key consideration lies in (1) decoupling the graph structure from the GNNs to account for the edge-specific modeling and (2) learning the graph structure from the global information in θ.

Towards the first purpose, the core idea is to decompose the graph structure information into connectedness (the edges in the adjacency matrix) and the strength of connection (the latent variable). In general, there are two ways to model the connection strength regarding whether relying on the inductive bias of translation invariant or not. On the one hand, attentive mechanisms or noise detection models are translation invariant, which decode the connection strength of each edge from its local features. However, with the consideration of the local structure heterogeneity issue in most real-world graphs [23], it is risky to rely on the translation invariant bias. On the other hand, probabilistic mechanisms separately model the connection strength for each edge, where each edge corresponds to a specific distribution. However, it is non-trivial to set a proper prior in practice. According to these advantages and disadvantages, we summarize two considerations for extending the graph convolution: (1) edge-specific modeling; and (2) optimization without prior.

Towards this end, we first propose the generic structure extractor (GSE) to decouple the graph structure into the connectedness between nodes and the strength of the connections. To model the edges in an edge-specific way, we apply the bi-level optimization method with the inner optimization to update the common parameters and the outer optimization to optimize the weight of each edge. By introducing the bi-level optimization, we can learn the structure information from the global parameters, thus avoiding the prior assumptions.

Specifically, we model the connection strength as a parameter matrix Z with the same size as A. Formally, the generic structure extractor (GSE) is abstracted as:$$\begin{array}{cc} \mathrm{GSE}(Z): & =\sigma \left(Z\right)⊙\tilde{A},\\ H^{\left(k\right)}=\mathrm{COM}(H^{\left(k-1\right)}, & \mathrm{AGG}(H^{\left(k-1\right)},\mathrm{GSE}\left(Z\right)), \end{array}$$
where σ(·) is a non-negative activation function, since the value of strength is always positive (in this work, we use the *min[max[0,x],1]* function to restrict the value within [0, 1]). **COM** and **AGG** are the combination and aggregation functions, respectively.

Noteworthy, different from GNNs, GSE decouples the graph structure from GNNs and treats it as a learnable objective. Besides, GSE is a generic extractor, which can be instantiated over most existing graph convolutions. In this way, as long as learning the connection strength is set properly, GSEBO can downweight the neighbors with inter-class connection during representation propagation, reducing the impact of inter-class with bi-level optimization.

### 3.3. Update Parameters with Bi-Level Optimization

To achieve the second purpose of learning graph structure from global information, it is essential to carefully design a proper training algorithm to optimize the connection strength matrix Z. Assume that we construct GSEBO with *K* layers, which is denoted as f(A,X;θ) with parameters $\theta =\{Z;W^{\left(k\right)}|k\in \left[1,K\right]\}$. We have three main considerations for designing the training algorithm:Connection strength is a relative value, which changes across different views. As shown in Figure 1b, the connection between *node 3* and *node 5* is weak from the local view, i.e., the edge is inter-class and should be assigned a low weight. However, this edge is essential for the classification of *node 9, 10, 11*, which deserves a high weight. Therefore, the optimization objective of Z should be the overall performance of node classification.Z and $W=\{W^{\left(k\right)}|k\in \left[1,K\right]\}$ play different roles, but are closely related. The role of Z is close to A, which restricts the model space for the mapping from node feature to label, and the role of W includes the global mapping information for classification, which would relieve the cons of local optimization. That is, an update on Z will adjust W and its optimization procedure. Therefore, the optimization of Z and W are at two different but dependent levels.Directly minimizing the objective function of Equation (Equation 1) to obtain the parameters Z and W is not able to achieve the desired purpose of learning the structure for the reason of over-fitting, which is shown in Figure 2.

Towards this end, we propose a bi-level optimization [24,25,26] with inner and outer optimizations to learn W and Z, respectively. Figure 3 shows the overall procedure of the algorithm, where the inner and outer optimization steps are iteratively executed until the stop condition.
(2)$$\min_{Z}F\left(W^{*}\left(Z\right)\right)=\sum_{v}l_{outer}\left(f{\left(A,X\right)}_{v},y_{v};W^{*}\left(Z\right)\right),$$
(3)$$\begin{array}{cc} s.t.\;W^{*}\left(Z\right) & \;=\;\arg\min_{W}L\left(W,Z\right)\\ & =\sum_{u}l_{inner}\left(f{\left(A,X\right)}_{u},y_{u};W,Z\right). \end{array}$$

#### 3.3.1. Inner Optimization for Common Parameters

This step is similar to the normal training of GNN for W optimization. In particular, we update W with a gradient descent based optimizer (e.g., Adam [27]) over the train_1 nodes by minimizing Equation (Equation 3) (i.e., set $l_{inner}=l\left(f\left(A,X\right),y;\theta \right)+\lambda ∥\theta ∥$). When calculating the gradient of W, we treat the connection strength matrix Z as constant.

#### 3.3.2. Outer Optimization for Graph Structure

Similarly, we treat the sequence of W as constant to optimize the graph structure Z. Ideally, the objective should be the overall classification loss of all nodes in graph. Formally:$$\begin{array}{c} \min_{Z}\sum_{i\leq N}l\left(f{\left(A,X\right)}_{i},y_{i};\theta \right). \end{array}$$

Apparently, the calculation of the ideal objective is intractable. Similar to the global parameters optimization, we can approximate the ideal objective as the empirical risk over train_1 nodes. However, it can easily suffer from the over-fitting issue, which is shown in Figure 2. We believe the empirical risk over train_2 nodes is a better approximation of the ideal objective since it reflects to what extent the parameter generalizes well. In this light, we set $l_{outer}$ as the classification loss and optimize Z over the train_2 set $V_{train2}$ by minimizing Equation (Equation 6).

Assuming:(4)$$F\left(W_{Z},Z\right)=\sum_{v\in V_{train2}}l\left(f_{W_{Z},Z}{\left(X,A\right)}_{v},y_{v}\right),$$

We optimize the outer parameters Z by fixing the inner parameters $W_{1},\cdots ,W_{\tau }$. Formally, the derivative of the outer objective to the hyperparameter Z (hypergradient) is formulated as: (5)$$\nabla_{Z}F\left(W^{*}\left(Z\right)\right)=\partial_{W}F\left(W^{*}\left(Z\right)\right)\nabla_{Z}W^{*}\left(Z\right)+\partial_{Z}F\left(W^{*}\left(Z\right)\right).\;Z=Z-\eta_{o}\nabla_{Z}F\left(W_{Z,\tau },Z\right).$$

For t=τ down to 1, we define the matrices:$$\begin{array}{cc} & \alpha_{t}=\left\{\begin{array}{c} \nabla F(W_{n})\;\;\;\;\;\;\;if\;n=\tau \\ \nabla F\left(W_{n}\right)M_{\tau }\cdots M_{n+1}\;if\;n=1,\cdots ,\tau -1 \end{array}\right. \end{array}$$
with:$$M_{n}=\frac{\partial W_{n}}{\partial W_{n-1}},\;N_{n}=\frac{\partial W_{n}}{\partial Z},$$
and the update of Z can be formulated as:(6)$$\begin{array}{cc} \nabla_{Z}F\left(W^{*}\left(Z\right)\right) & =\nabla_{Z}F\left(W_{\tau }\right)\sum_{r=1}^{\tau }\left(\prod_{s=r+1}^{\tau }M_{s}\right)N_{r},\\ Z=Z & -\eta_{o}\nabla_{Z}F\left(W^{*}\left(Z\right)\right). \end{array}$$

#### 3.3.3. Training Process

The bi-level optimization cannot guarantee the convergence of the model; thus, we set the early stop condition as the additional requirement for training, which is shown in the second line of Algorithm 1. We set the early stop condition as: within 20 epochs, if the accuracy of the verification set is not improved, the model will stop training.
**Algorithm 1** Training of GSEBO.**Require:**A: adjacency matrix; X: nodes features; $\eta_{o},\eta_{i}$: outer and inner learning rates; τ:  number of inner steps;
1:Initialize W and Z (We initialize Z with the normalized $\tilde{A}$.);2:**while** not *early stopping* **do**3:    **for** t=1 to τ **do**              # inner optimization;4:        Update W on train_1 dataset *w.r.t.* Equation (Equation 1);5:    **end for**6:    $\alpha_{\tau }=\nabla_{Z}\sum_{v\in V_{train2}}l\left(f{\left(A,X\right)}_{v},y_{v};W_{\tau }\right)$;7:    P=0;                      # initial outer gradient;8:    **for** t=τ−1 downto 1 **do** # calculate outer gradient;9:        $M_{t+1}=\frac{\partial W_{t+1}}{\partial W_{t}}$, $N_{t+1}=\frac{\partial W_{t+1}}{\partial Z}$;10:        $P=P+\alpha_{t+1}N_{t+1}$;11:        $\alpha_{t}=\alpha_{t+1}M_{t+1}$;12:    **end for**13:    Update $Z=Z-\eta_{o}P$;        # outer optimization;14:**end while**

Assuming that τ times of gradient descent, an approximate solution $W_{1},\cdots ,W_{\tau }$ of the inner optimization problem are obtained. For t=1 to τ, the updated W is calculated as $W_{t}=W_{t-1}-\eta_{i}\nabla L\left(W_{t-1},Z\right)$, where $\eta_{i}$ is the inner learning rate, and the process of updating W is shown on line 3–5 in Algorithm 1.

The process of updating Z is shown on line 8–11 in Algorithm 1. With the parameter Z updated by the graph structure optimization, we reoptimize the W with Z by global parameters optimization, and repeat this process until the early stopping is met. To summarize, Algorithm 1 shows the training procedure of GSEBO.

#### 3.3.4. Complexity Analysis

The overall framework of GSEBO is illustrated in Algorithm 1. The computing complexity of our GSEBO mainly depends on two steps. Given a graph *G*, ${\left||A|\right|}_{0}$ is the number of nonzeros in the adjacency matrix, *d* is the feature dimension, *L* is the layer number of GCN, and |V| is the number of nodes. In the inner optimization step, the graph convolutional network takes $O(\tau L||A|{|}_{0}d)$ computational time. In the outer optimization step, the computational time is $O(\tau ||A|{|}_{0}^{2})$. As a result, the total computational is $O(\tau ||A|{|}_{0}^{2})$ if Ld is smaller than ${\left||A|\right|}_{0}$.

### 3.4. Unifying Various GNN and Beyond

To extend the graph convolution operator to the most existing graph convolutions, we reformulate the classical GNN model into a unified form.

#### 3.4.1. Gcn

The forward propagation of GCN is formulated as follows:$$H^{\left(l\right)}=\sigma \left({\tilde{D}}^{-\frac{1}{2}}\tilde{A}{\tilde{D}}^{-\frac{1}{2}}H^{\left(l-1\right)}W\right),$$
where $H^{\left(0\right)}=X$, $\tilde{A}=A+I$, ${\tilde{D}}_{nn}=\sum_{i}{\tilde{A}}_{ni}$, and $W\in R^{D\times C}$ are the trainable parameters, I denotes the identity matrix, and σ is a nonlinear function. For better understanding of the propagation of GCN, we rewrite the embedding update in the following form:$$\begin{array}{c} h_{v}^{\left(l\right)}=\sum_{u\in N(v)+v}\frac{1}{\sqrt{d_{v}}\sqrt{d_{u}}}W^{\left(l-1\right)}h_{v}^{\left(l-1\right)},\\ H{}^{\left(l\right)}=\sigma_{1}{\left(\sigma_{2}\left(Z\right)⊙\left(\tilde{A}\right)\right)}^{\left(l-1\right)}H^{\left(l-1\right)}W^{\left(l-1\right)}),\\ Z_{ij}=\frac{1}{\sqrt{d_{i}}\sqrt{d_{j}}}\;\;\;where\;\;{\tilde{A}}_{ij}=1,\;and\;j\in N\left(i\right), \end{array}$$
where $W^{\left(l\right)}$ reflects the trainable parameters on the *l*-th layer, $\sigma_{1}$ is the activate function, such as ReLU, and $\sigma_{2}=\min\left[\max\left[0,x\right],1\right]$ is the a non-negative activation function.

#### 3.4.2. Gat

GAT assigns different weights to each neighbor node, and updates the node embeddings with weighted average of neighbors:$$\begin{array}{c} H^{\left(l\right)}=\sigma_{1}\left(\frac{1}{K}\sum_{k=1}^{K}\left(\left(\sigma_{2}\left(Z_{k}\right)⊙\tilde{A}\right)H^{\left(l-1\right)}W_{k}^{\left(l-1\right)}\right)\right),\\ Z_{k,ij}=Softmax(atten\left(W_{k}^{\left(l-1\right)}h_{i}^{\left(l-1\right)},W_{k}^{\left(l-1\right)}h_{j}^{\left(l-1\right)}\right))\\ where\;\;{\tilde{A}}_{ij}=1,\;and\;j\in N\left(i\right), \end{array}$$
where $W_{k}^{\left(l-1\right)}$ is the parameter of *k*th multi-head attention on (l−1)th layer, Softmax is the Softmax operation, and atten is the self-attention operation.

#### 3.4.3. GraphSAGE

GraphSAGE is one representative of the spatial approaches, which learns how to aggregate feature information from a node’s local neighborhood, and the reformulation is as follows:$$\begin{array}{c} H^{\left(l\right)}=\sigma_{1}(Concat\left(H^{\left(l-1\right)},\left(\sigma_{2}\left(Z\right)⊙A\right)H^{\left(l-1\right)})W^{\left(l-1\right)}\right),\\ Z_{ij}^{\left(l-1\right)}=\frac{1}{d_{i}}\;\;\;where\;\;A_{ij}=1,\;and\;j\in N\left(i\right). \end{array}$$

#### 3.4.4. JK-Net

JK-Net learns the node representation from deep layers by using a jumping network. There exist different ways to aggregate features from different layers; we take the concatenation as an example in our work (other aggregation methods are also applicable). The formulation of JK-Net is as follows: $$\begin{array}{c} H^{\left(L\right)}=FC(Concat\left(H^{\left(0\right)},H^{\left(1\right)},\cdots ,H^{\left(L-1\right)}\right))\\ with\;H^{\left(l\right)}=\sigma_{1}(\left(\sigma_{2}\left(Z\right)⊙\tilde{A}\right)H^{\left(l-1\right)}W^{\left(l\right)},\\ Z_{ij}=\frac{1}{\sqrt{d_{i}d_{j}}}\;\;\;where\;\;{\tilde{A}}_{ij}=1,\;and\;j\in N\left(i\right), \end{array}$$
where FC is a fully connected layer and Concat is the concatenation operation.

## 4. Experiments

In this section, we conduct experiments on four datasets to answer the following research questions:**RQ1:** How does the performance of the proposed GSEBO compared with the state-of-the-art methods?**RQ2:** How robust is the proposed GSEBO under different noisy levels?**RQ3:** To what extent does the proposed GSEBO decrease the impact of inter-class connections?**RQ4:** What are the factors that influence the effectiveness of the proposed GSEBO?

### 4.1. Experimental Setup

#### 4.1.1. Dataset

We select four widely used real-world node classification benchmark datasets with graphs of citation networks (Cora and Citeseer [21]), social networks (Terrorist) [28], and air traffic (Air-USA) [29]. The statistics of the datasets are shown in Table 1.

**Citation networks.** Cora and Citeseer are citation networks, where the nodes are papers published in computer science, the features of each publication are described by a 0/1-valued word vector indicating the absence/presence of the corresponding word from the dictionary, the adjacency matrix is binary and undirected, which denotes the citation relation between papers, and the labels are the category of each paper.**Terrorist Attacks.** Terrorist attacks describe the information related to terrorism attack entities: the attributes of the entities and the links that connect various entities together to form a graph structure. This dataset consists of 1293 terrorist attacks each assigned one of six labels indicating the type of the attack. Each attack is described as a 106-dimensional vector, with a 0 and 1 value indicating the absence and presence, respectively, of an entity. The edges are binary and undirected.**Air traffic network.** Air-USA is the airport traffic network in the USA, where the nodes represent airports and the binary undirected edges indicate the existence of commercial flights between the airports. The features of nodes are one-hot degree vectors and the labels are generated based on the activity measured by people and flights passed the airports.

All the baselines and our proposed method can be applied to all types of networks. We adopt the same data split of Cora and Citeseer as [21], and a split of training, validation, and testing with a ratio of 10:20:70 on other datasets [12].

#### 4.1.2. Compared Methods

We apply GSEBO on four representative GNN architectures: GCN [21], GAT [5], GraphSAGE [22], and JK-Net [23]. For each GNN, we compare GSEBO with the vanilla version, and three variants with state-of-the-art connection modeling methods: AdaEdge [30], DropEdge [31], and GAUG [12]. In addition, we compared the GSEBO with GSL methods: BGCN [17], VGCN [18], PTDNet [13], and advanced attention mechanism: MAGNA [6]. Note that the base model of GSEBO, BGCN, and PTDNet is GCN.

#### 4.1.3. Implementation Details

In the experiments, the separate ratio of training data is set to 0.8 for the train_1 and train_2 datasets, and we optimize the outer and inner parameters on the train_1 and train_2 datasets, respectively. The latent dimension of all the methods is set to 16. The parameters for all baseline methods are initialized as the corresponding papers, and are carefully tuned to achieve optimal performances. The learning rate of inner and outer optimization is set to 0.01. The hyperparameter λ is set to $5\times 10^{-4}$, and we search for the inner learning depth τ over the range [5,10,15,20,25]. As bi-level optimization cannot guarantee the convergence, we set the patience of early stopping to 20 to terminate the optimization of GSEBO. To prevent overfitting, the dropout ratio is set to 0.5 for all the methods. The network architectures of all the methods are configured to be the same as described in the original papers. Our experiments are conducted with Tensorflow running on GPU machines (NVIDIA 2080Ti). For all the compared methods, we report the average accuracy on the test set over 10 runs.

### 4.2. Performance Comparison

Table 2 shows the performance of GSEBO instantiate with classical GNNs, and Table 3 presents the results comparison of GSEBO and state-of-the-art GSL methods. From Table 2 and Table 3, we have the following observations:

#### 4.2.1. Improvement over Baselines

GSEBO outperforms the baselines in most cases. Considering the average performance over four datasets, the improvement of GSEBO over the baselines is in the range of 3.0–12.5%, which validates the effectiveness of the proposed method. In particular:**Probabilistic mechanisms**. The performance gain of BGCN and VGCN over the vanilla GCN are limited, which might because of the unsatisfied assumption of the prior distribution. This result shows the rationality of relaxing the prior assumption and modeling the connection strength with parameters directly.**Connection modeling**. DropEdge, AdaEdge, GAUG, and PTDNet achieve better performance than vanilla GNN, which modifies the structure of the graph from different perspectives, such as the smoothness [30] and robustness [31]. These results reflect the benefit of connection modeling. However, there is still a clear gap between these methods and GSEBO, which is attributed to the global objective for learning the edge strength.**Attention mechanism**. The performance of MAGNA is inferior, which is consistent with the analysis in [10].

#### 4.2.2. Effects across GNN Architectures

On the four GNN architectures, GSEBO achieves better performance than the vanilla version in all cases. In particular, GSEBO achieves an average improvement across datasets of 4.2%, 4.4%, 4.0%, and 5.74% over the vanilla GCN, GAT, GraphSAGE, and JK-Net, respectively. These results justify the effectiveness of structure learning of the proposed GSEBO. Across the four architectures, GSEBO achieves the largest improvement over JK-Net. We postulate the reason is that the jump connection in JK-Net makes it aggregate more hops of neighbors than the other GNNs. As the hops increase, the homophily ratio of the neighbors will decrease, i.e., more neighbors are in classes different from the target node. Therefore, optimizing the connection strength (i.e., Z) is more beneficial on JK-Net.

#### 4.2.3. Effects across Datasets

From the perspective of the dataset, GSEBO consistently performs better than the vanilla version on the four datasets. Specifically, the average improvement over the four across classic GNN architectures achieved by GSEBO are 1.9%, 3.1%, 5.3%, and 8.0% on the four datasets, respectively. Considering that the four datasets come from different scenarios, these results are evidence for the potential of GSEBO to be widely applied in practice. Note that the trend of performance improvement is similar to the density of graph, where Air-USA is the most dense graph with the largest performance improvement. As the number of neighbors increases, the percentage of neighbors essential for the classification of the target node will decrease. This result can reflect the rationality of optimizing the connection strength according to the overall classification objective.

Moreover, the training loss and test accuracy of GNN methods and the proposed GSEBO are shown in Figure 4. We observe that the loss of GSEBO tend to be stable after 30 epochs and is smaller than GCN, GAT, and GraphSAGE, showing the empirical convergence of GSEBO. This is because simultaneously optimizing the parameters of W and Z would make the loss function more easily approach to the labels. However, the loss of JK-Net is smaller than GSEBO, which we attribute to that JK-Net is easier to trap in over-fitting.

### 4.3. Robustness Analysis

We investigate the robustness of GSEBO under the different inter-class levels. In particular, we follow [13] and construct graphs based on Cora by randomly adding 1000, 3000, 5000, 10,000, and 20,000 inter-class edges, respectively. On the synthetic datasets, we compare GSEBO with Vanilla, GAUG, DropEdge, and AdaEdge. Figure 5 shows the performance on the five GNN architectures. From the figures, we have the following observations:The margins between GSEBO and vanilla GNN on the synthetic datasets are larger than the original Cora. For example, when adding 20,000 inter-class edges, GSEBO improves the accuracy by 17.6%, 3.1%, 40.6%, and 4.4% compared to GCN, GAT, GraphSAGE, and JKNet. This result indicates the robustness of GSEBO’s structure learning ability.In most cases, GSEBO outperforms the baselines at different noisy levels, which further justifies its robustness.GAUG and AdaEdge utilizes different strategies to update the structure of the graph, which also consistently perform better than vanilla GNN. However, their gaps to GSEBO on the synthetic data are larger than the original Cora. We postulate that the reason for this is that their objectives are affected by the intentionally added noise.DropEdge shows worse performances than the vanilla GNN on the synthetic datasets. The comparison shows that randomly dropping edges fails to enhance GNN robustness when the noisy level is high.

### 4.4. Visualization of Denoising (RQ3)

To investigate the effect of the connection strength matrix Z, we further visually compare the initialization of Z (i.e., initial weight) and its value after optimization (i.e., denoised weight). As both of them are large and sparse matrix, we visualize part of the denoised weight with a region of 100×100, which is shown on Figure 6. We show the results on Core and Air-USA, where the improvement of GSEBO is the smallest and largest, respectively. In the figures, we use red pixels and blue pixels to represent intra-class and inter-class edges, respectively, where a deeper color represents larger values. From Figure 6a,b, we find that the denoised weight matrix mainly decreases the value corresponding to inter-class connections, which, thus, can downweight the corresponds neighbor during representation propagation. In some cases, the denoised weight increases the weight of intra-class connections, which also facilitates denoising. Apparently, most edges in Air-USA are intra-class, and GSEBO still downweights a portion of inter-class edges and increases the intra-class weights, as shown in Figure 6c,d. This is because the denoised weight is learned in order to reach the overall classification objective.

### 4.5. Hyperparametric Analysis (RQ4)

We then investigate how the hyperparameters of GSEBO affect its effectiveness. As introduced in Section 3, the most critical hyperparameter of GSEBO is the training depth of inner optimization step (i.e., τ), which also is the number of gradient update in each inner optimization step. For the consideration of computing cost, we cap the value of τ at 25 and changes its value with step of 5, i.e., setting τ∈{5,10,15,20,25}.

Figure 7 shows the performance of GSEBO applied to the four classical GNN architectures on the four datasets as changing the value of τ. From the figures, we have the following observations: (1) in most cases, the overall performance of the proposed GSEBO on each dataset is relatively stable as increasing the value of τ. This result indicates the insensitivity of GSEBO to the hyperparameter. (2) While increasing the value of τ can slightly improve the performance of GSEBO, we still suggest setting it as a relatively small value. This is because enlarging τ will increase the memory and computation cost of GSEBO.

## 5. Conclusions

In this work, we propose a novel GSEBO, which utilizes the generic structure extractor to extract the graph structure as learnable parameters and learns the structure from the global view. To better optimize the structure and common parameters, we decompose it into two mutually constrained optimization objectives, i.e., a bi-level optimization, with inner and outer optimization for common parameters and graph structure optimization, respectively. Extensive experiments demonstrate the effectiveness and robustness of GSEBO on both benchmark and synthetic datasets.

Even though GSEBO achieved impressive results, there are still some limitations: GSEBO cannot be effectively applied to large graphs, which requires implementation of mini-batches. Besides, we evaluate GSEBO in the transductive setting; when new nodes are added to the graph after training, GSEBO has to retrain the entire model. For future research, we would like to explore solutions to the above limitations.

## Figures and Tables

**Figure 1 entropy-24-01228-f001:**
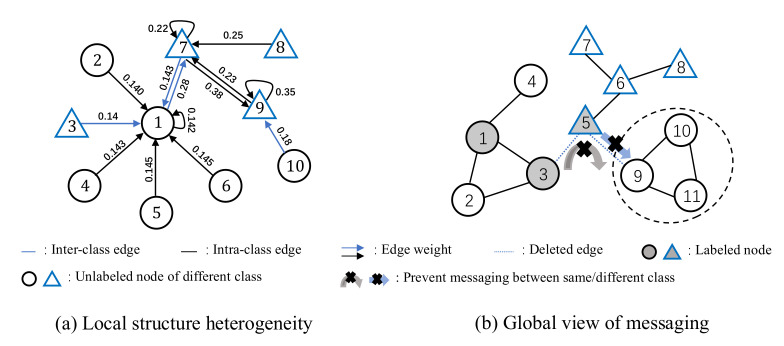
Examples for (**a**) the impact of local structure heterogeneity (*node 1* and *9*, the weights between nodes do not provide enough information for classes differentiation) and (**b**) the global view of messaging (the edge between *node 3* and *node 5*, the deleted edge prevents the intra-class information transfer).

**Figure 2 entropy-24-01228-f002:**
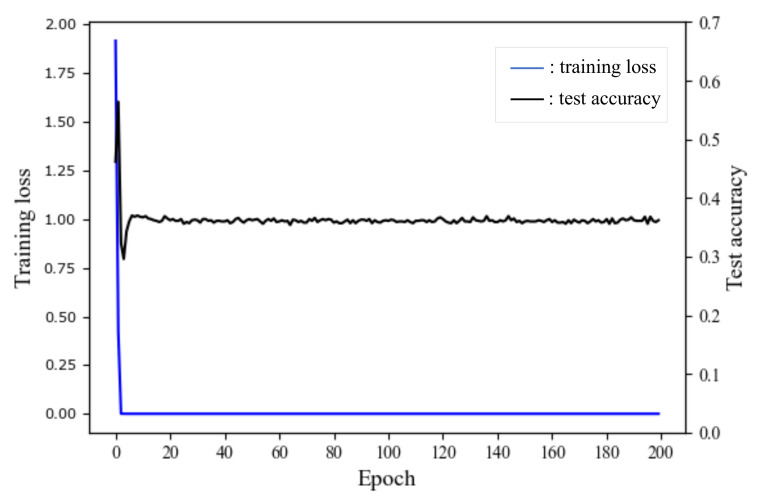
Optimize inner and outer steps on the training set of Cora.

**Figure 3 entropy-24-01228-f003:**
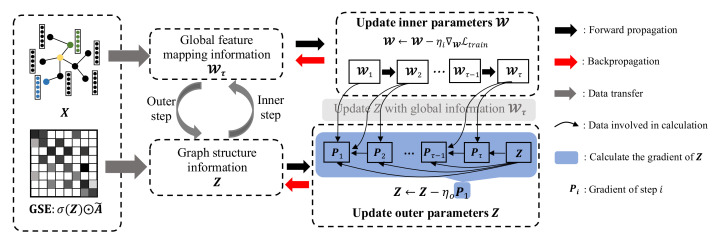
The framework of GSEBO. The generic structure extractor (GSE) decouples the graph structure from GNNs to account for the edge-specific modeling; the inner and outer optimization steps are adopted for common parameters and graph structure optimization.

**Figure 4 entropy-24-01228-f004:**
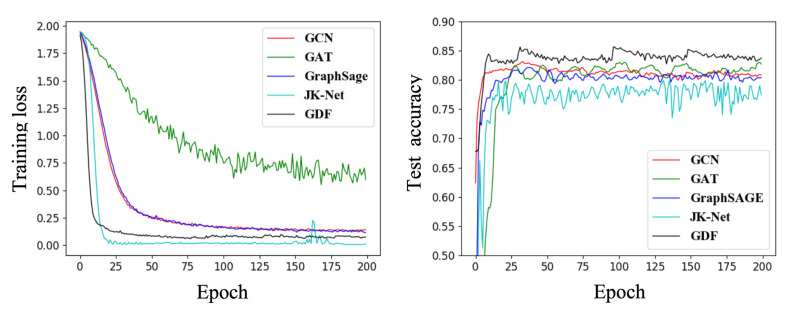
The training loss and test accuracy of classical GNN and GSEBO on Cora.

**Figure 5 entropy-24-01228-f005:**
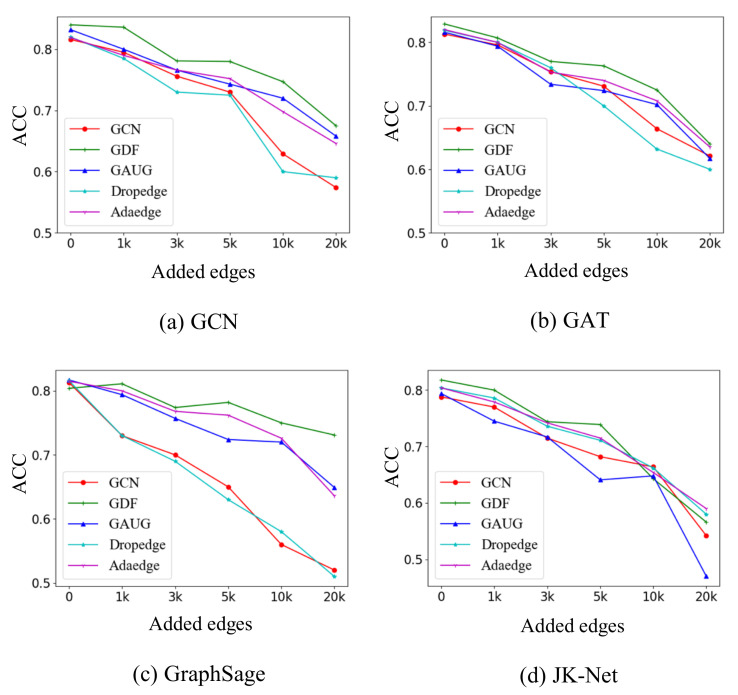
Node classification performance of GSEBO on Cora poisoned at different inter-class levels.

**Figure 6 entropy-24-01228-f006:**
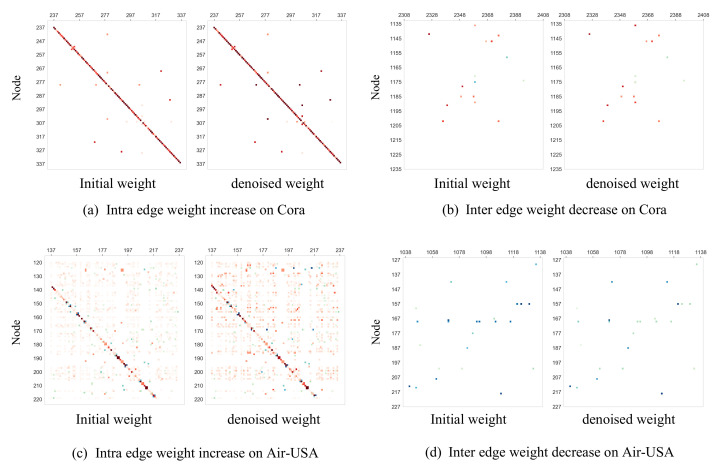
Comparison of the initial weight and denoised weight to visualize the effect of denoising.

**Figure 7 entropy-24-01228-f007:**
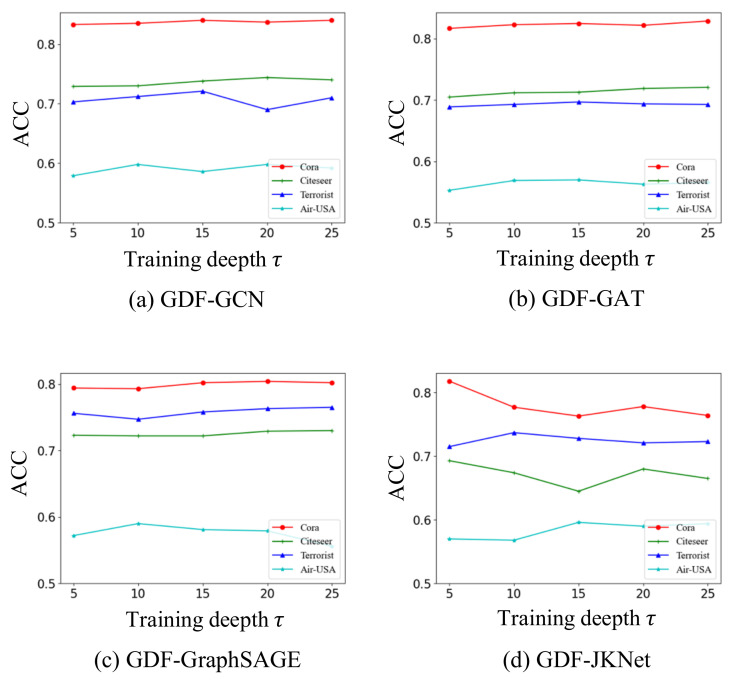
The performance of GSEBO on the four datasets as changing the value of τ from 5 to 25.

**Table 1 entropy-24-01228-t001:** Summary statistics for the datasets.

	Cora	Citeseer	Terrorist	Air-USA
Nodes	2708	3327	1293	1190
Edges	5278	4552	3172	13,599
Inter ratio	0.151	0.194	0.362	0.289
Features	1433	3703	106	238
Class	7	6	6	4
Training set	140	120	129	119
Validation set	500	500	258	238
Testing set	1000	1000	906	833

**Table 2 entropy-24-01228-t002:** Performance comparison across GNN architectures.

Method	Cora	Citeseer	Terrorist	Air-USA
GCN				
Vanilla	81.6 ± 0.7	71.6 ± 0.4	70.0 ± 1.1	56.0 ± 0.8
AdaEdge	81.9 ± 0.7	72.8 ± 0.7	71.0 ± 1.9	57.2 ± 0.8
DropEdge	82.0 ± 0.8	71.8 ± 0.2	70.3 ± 0.9	56.9 ± 0.6
GAUG	83.2 ± 0.7	73.0 ± 0.8	71.4 ± 2.0	57.9 ± 0.4
GSEBO	84.1±0.5	74.3±0.4	72.3±0.8	59.6±0.5
GAT				
Vanilla	81.3 ± 1.1	70.5 ± 0.7	67.3 ± 0.7	52.0 ± 1.3
AdaEdge	82.0 ± 0.6	71.1 ± 0.8	72.2±1.4	54.5 ± 1.9
DropEdge	81.9 ± 0.6	71.0 ± 0.5	69.9 ± 1.1	52.8 ± 1.7
GAUG	81.6 ± 0.8	69.9 ± 1.4	68.8 ± 1.1	53.0 ± 2.0
GSEBO	82.8±0.3	72.2±0.9	69.4 ± 1.2	57.3±0.8
GraphSAGE				
Vanilla	81.3 ± 0.5	70.6 ± 0.5	69.3 ± 1.0	57.0 ± 0.7
AdaEdge	81.5 ± 0.6	71.3 ± 0.8	72.0 ± 1.8	57.1 ± 0.5
DropEdge	81.6 ± 0.5	70.8 ± 0.5	70.1 ± 0.8	57.1 ± 0.5
GAUG	81.7±0.3	71.4 ± 1.0	70.4 ± 0.5	55.0 ± 1.1
GSEBO	80.7 ± 0.8	73.1±0.4	76.4±0.9	59.2±1.1
JK-Net				
Vanilla	78.8 ± 1.5	67.6 ± 1.8	70.7 ± 0.7	53.1 ± 0.8
AdaEdge	80.4 ± 1.4	68.9 ± 1.2	71.2 ± 0.7	59.4 ± 1.0
DropEdge	80.4 ± 0.7	69.4 ± 1.1	70.2 ± 1.3	58.9 ± 1.4
GAUG	79.4 ± 1.3	68.9 ± 1.3	70.2 ± 0.5	52.3 ± 1.8
GSEBO	81.6±1.1	69.5±1.4	73.4±1.4	59.8±1.1

**Table 3 entropy-24-01228-t003:** Performancecomparison of GSEBO with GSL methods.

Method	Cora	Citeseer	Terrorist	Air-USA
BGCN	81.2 ± 0.8	72.4 ± 0.5	70.3 ± 0.8	56.5 ± 0.9
VGCN	64.4 ± 0.2	67.8 ± 0.8	73.8±0.9	53.3 ± 0.3
PTDNet	82.8 ± 2.6	72.7 ± 1.8	68.3 ± 1.6	53.4 ± 1.4
MAGNA	81.7 ± 0.4	66.4 ± 0.1	67.2 ± 0.1	55.1 ± 1.2
GSEBO	84.1±0.5	74.3±0.4	72.3 ± 0.8	59.6±0.5

## Data Availability

The data presented in this study are openly available in an open access repository at [21,28,29].

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
