# Peer review of "Generic Structure Extraction with Bi-Level Optimization for Graph Structure Learning"

_entropy, 2022, doi:10.3390/e24091228_

Round 1

Reviewer 1 Report

In their manuscript, the authors propose an innovative Graph Structure Learning (GSL) method, based on a Bi-Level Optimization. The key idea is to optimize the parameters of a Graph Neural Network (GNN) in problems of node classifications, acting on two levels: an upper one, that learns the graph structure in a global form, optimizing a global node classification objective, and a lower one, in which the graph structure is refined considering the information collected at the previous stage. The proposed method is tested on consolidated GNN architectures, and the results are compared with state-of-the-art GSL methods. Validation is performed by considering four real-world networks.

The research is based on an innovative and robust methodology, clearly and accurately explained throughout the manuscript. Therefore, I would recommend that the paper is considered for publication, provided the following minor issues are solved:

1.     In Section 3.1, the same letter (M) is used to denote two different quantities: the number of edges in the network and the number of known node labels. For consistency, one of the two symbols must be modified.

2.      In this version, Figure 2 is shown way before the text reference to it. Moreover, the text reference to Figure 2 appears even later than the one to Figure 3. In my opinion, the order of the two figures must not be changed, since Figure 2 shows the general scheme of the workflow proposed in the manuscript. However, its text reference must be appropriately placed (e.g., in Section 3.4).

3.     In the description of real-world networks used for model validation, the authors should specify further information. In particular, it is not clear how the edges in the terrorist attacks network are built, and what do the 106 mentioned features represent. Clearly, it is not necessary to list them, but at least to specify to which property are they referred. In general, the authors must specify if real-world networks are either binary or weighted, and either directed or undirected. It is also worth specifying if the method they discuss can be applied or not to any type of network.

4.     The authors should check the information reported at lines 243-244: is it possible that only 10% of data is used in the training set?

5.     I would suggest to explain in more detail the Denoising procedure discussed in Section 4.4.

6.     Some typos must be corrected throughout the manuscript. Here I report some of them: at line 165, “activate function” should read “activation functions”; at line 167, “update” should read “updates”; at line 172, “fully connect layer” should read “fully connected layer”; at line 229 “the features of each publication is described” should read “the features of each publication are described”; at line 267, “Tabel 3” should read “Table 3”; at line 340, “read” should read “red”; at lines 342-343, “corresponds” should read “corresponding”.

7.     The analysis is based on implementing a graph structure extractor, used after decomposing information on the network in two elements: connectivity, represented through the adjacency matrix, and connection strength, which constitute the latent variable. It would be interesting if the authors comment, even as a matter of interest for future perspective, if the same approach can be employed using the normalized graph Laplacian instead of the adjacency matrix. Actually, diagonalization of the normalized Laplacian provides the graph spectrum, an established and well-known tool to characterize complex networks, representing in a concise way information on connectivity and mesoscale network organization, as discussed, e.g., in the following references: William N. Anderson Jr. & Thomas D. Morley, Eigenvalues of the Laplacian of a graph, Linear and Multilinear Algebra, 18, 141-145, (1985) https://doi.org/10.1080/03081088508817681; Amoroso, N., Bellantuono, L., Pascazio, S. et al. Potential energy of complex networks: a quantum mechanical perspective. Sci Rep 10, 18387 (2020) https://doi.org/10.1038/s41598-020-75147-w; Braunstein, S.L., Ghosh, S. & Severini, S. The Laplacian of a Graph as a Density Matrix: A Basic Combinatorial Approach to Separability of Mixed States. Ann. Comb. 10, 291–317 (2006) https://doi.org/10.1007/s00026-006-0289-3 ; Vukadinovic, D., Huang, P. & Erlebach, T. A spectral analysis of the Internet topology. ETH TIK-NR 118, 1–11 (2001).

Reviewer 2 Report

Dear authors, 

Thanks for submitting to the journal. I enjoyed reading the paper and the proposed approach. I was unclear on a few proposed ideas and would advise correcting them before the camera-ready draft. 

Summary: The paper proposes Generic Structure Extraction with Bi-level Optimization for Graph Structure Learning (GSEBO) to address the limitations in the current state-of-the-art graph structure learning models. The current state-of-the-art methods learn by computing local information about a given edge and fail to capture global information resulting from "local structure heterogeneity." (local structure heterogeneity occurs when weights between nodes do not have enough information to differentiate output label classes). The authors hypothesize that this lack of information capture can reduce the model's accuracy when performing node classification tasks. GSEBO proposes a two-step solution capturing a global view by optimizing a global objection of node classification task to address this. In the first step, a generic graph structure extracts an abstract graph structure (basically an adjacency matrix) in the form of a learnable variable Z that represents the edge's strength between nodes of interest. In the second step, the authors propose a loss function that independently learns both Z and traditional weight matrix W as a two-step learning process. Using four GNN models and small sizes graph dataset, the authors validate the approach and its results. 

Improvements/Questions: 

  1. Not having enough labels to train well is a valid concern in state-of-the-art models. However, the proposed approach does not seem to tackle this problem. Please acknowledge this as a limitation. 
  2. Line 151 and 152 are vital parts of the paper and are explained poorly. It is still unclear how these two considerations: "edge specific modeling and optimization without a prior," matter and how in the proposed approach it has overcome. Please expand on this with additional two paragraphs. 
  3. Parameter matrix Z is a generalization of the GAT approach. Is that the correct understanding? If not, what am I missing?
  4. The size of the parameter matrix Z is vast and can significantly hamper training. Please add a section describing the memory complexity and capacity required or increased by the proposed model. 
  5. Likewise, the two-step training does not guarantee convergence, and because of additional learnable parameters, it also increases the computation burden. Please add a table or section covering the additional requirement for training the model. 
  6.  The paper does not explain why one should initialize Z with the normalized A^. Please add a sentence for this. 
  7. Line 262 - mehtods --> methods.
